# Head-Out Water-Based Protocols to Assess Cardiorespiratory Fitness—Systematic Review

**DOI:** 10.3390/ijerph17197215

**Published:** 2020-10-02

**Authors:** Anna Ogonowska-Slodownik, Paula Richley Geigle, Natalia Morgulec-Adamowicz

**Affiliations:** 1Faculty of Rehabilitation, Jozef Pilsudski University of Physical Education in Warsaw, Marymoncka 34, 00-968 Warszawa, Poland; natalia.morgulec@awf.edu.pl; 2Department Physical Therapy, South College, Knoxville, TN 37922, USA; geigle@comcast.net

**Keywords:** aerobic capacity, oxygen consumption, exercise, testing, aquatic

## Abstract

The aquatic environment offers cardiorespiratory training and testing options particularly for individuals unable to adequately train or test on land because of weight bearing, pain or disability concerns. No systematic review exists describing cardiorespiratory fitness protocols used in an aquatic environment. This review investigated the different head-out water-based protocols used to assess cardiorespiratory fitness. Our comprehensive, systematic review included 41 studies with each included paper methodological quality assessed using the statistical review of general papers checklist. Diverse protocols arose with three major categories identified: conducted in shallow water, deep water, and using special equipment. Thirty-seven articles presented data for peak/maximal oxygen consumption (VO_2peak_/VO_2max_). Twenty-eight of 37 studies predefined criteria for reaching a valid VO_2peak_/VO_2max_ with shallow water exercise demonstrating 20.6 to 57.2 mL/kg/min; deep water running 20.32 to 48.4 mL/kg/min; and underwater treadmill and cycling 28.64 to 62.2 mL/kg/min. No single, accepted head-out water-based protocol for evaluating cardiorespiratory fitness arose. For clinical use three cardiorespiratory fitness testing concepts ensued: water temperature of 28–30 °C with difference of maximum 1 °C between testing participants and/or testing sessions; water depth adapted for participant aquatic experiences and abilities; and intensity increment of 10–15 metronome beats per minute.

## 1. Introduction

Cardiorespiratory fitness (CRF) is a health-related component of physical fitness defined as the ability of the circulatory, respiratory, and muscular systems to supply oxygen during sustained physical activity [1]. CRF is not only a sensitive and reliable measure of habitual physical activity [2] but also a relatively low-cost and useful client health indicator in clinical practice [3]. Therefore, it is increasingly important to monitor and systematically evaluate cardiorespiratory fitness in persons with disability or chronic disease in clinical rehabilitation and beyond [4,5,6]. Monitoring changes in cardiorespiratory fitness may indicate training and rehabilitation program effectiveness, as well as the development of a physically active lifestyle [7].

Water offers a place for cardiorespiratory training and testing especially for populations who cannot be trained or tested on land because of problems with weight bearing, pain or disability. The head-out aquatic exercises are an important therapeutic component for individuals with physical limitations [8,9], as well as an element of a primary health prevention system, [10,11,12] and elite athlete sport performance conditioning [13,14,15,16,17]. Regular testing of the individuals participating in an aquatic therapy program with standardized cardiorespiratory protocols may give practitioners valuable information for establishing exercise guidelines, monitoring progress and making adjustments in both the intervention and training program. Although various cardiorespiratory fitness protocols used in the aquatic environment are available, the disparity in testing protocol proprieties (e.g., water temperature and depth, water immersion level, starting intensity, stage time, equipment used, test termination criteria, criteria for reaching a valid VO_2peak_/VO_2max_) and outcomes often hampers the process of data integration and interpretation. Additionally, practitioners struggle to compare trends across studies, with generalization to the larger population impeded. Given the importance of cardiorespiratory fitness assessment in a water environment, more standardized guidelines for testing are required.

Additionally, understanding and evaluating specific aquatic exercise/therapy prescription outcomes necessitates parallel treatment and evaluation environments. Body posture during immersion is significant [18]. Head-out water immersion increases central blood volume and minimizes the influence of gravity. Cardiovascular and metabolic adaptations are major interest areas in head-out aquatic exercises, since they are related to modulating pathologies such as coronary artery disease, hypertension, stroke, obesity and diabetes [8]. Evidence from water-based studies indicates regular deep or shallow water exercise can exert beneficial effects on cardiorespiratory fitness, strength, and body fat distribution [10]. Cardiovascular function improvement can be explained by the elevation in cardiac output due to the blood flow shift to the main blood vessels while immersed [19]. The hydrodynamic properties of water offer safety, facilitating training and testing for individuals with compromised musculoskeletal, neurologic, and cardiorespiratory conditions [20,21,22]. For example, head-out aquatic exercise is an effective training and conditioning method for postmenopausal women to improve strength, flexibility, functional movements, bone density, and quality of life [20].

Cardiopulmonary exercise testing is regarded as the criterion for clinical procedure and the optimal method for assessing cardiorespiratory fitness by quantifying peak oxygen consumption (VO_2peak_), which represents an individuals’ capacity to produce energy to perform strenuous exercise [23]. Reliable and valid cardiorespiratory testing requires consistent procedures and optimal measurement processes [24]. No systematic review describes currently available cardiorespiratory fitness protocols used in the aquatic environment. This review explored published head-out water-based protocols and identified specific cardiorespiratory fitness testing properties used in aquatic environment. The aims of this systematic review were: (I) to summarize the available head-out water-based protocols; (II) to identify specific testing proprieties for cardiorespiratory fitness protocols used in the aquatic environment; and (III) to analyze cardiorespiratory fitness values and population characteristics. Based on this review, suggestions and recommendations for clinical and practical use and continued research were provided.

## 2. Methods

### 2.1. Search Strategy and Study Selection

The protocol for systematic review was registered in PROSPERO (CRD42020159400). For this systematic review we performed search of these databases: Embase, Cinahl, SPORTDiscus, PubMed, and Medline using a comprehensive combination of keywords. The search included publications in English, appearing before 31st November 2019. The keywords were (Oxygen consumption OR Oxygen uptake OR Cardiorespiratory fitness OR Cardiorespiratory response OR Aerobic capacity OR Aerobic fitness) AND (Aqua OR Aquatic OR Water-based) AND (Exercise OR Testing OR Test OR Protocol). An example of search parameters, for Pubmed, can be found in Appendix A.

Duplicates were removed using Endnote (Endbase X8, Thomson Reuters, New York, NY, USA). Based on title and abstract non-relevant studies were excluded. Exclusion criteria were: animal studies, not head-out water-based protocols, head-out water-based protocols not measuring cardiorespiratory fitness, articles with intervention in aquatic setting without testing cardiorespiratory fitness, reviews of the literature, articles not in English, and articles without full text.

Inclusion of each paper was based on the assessment of two independent reviewers (AOS and NMA) and full agreement was required. Studies included in our review were published in English and fulfilled the following criteria according to the PICOS system (Population: human; Intervention: not applicable, Comparison: other protocols assessing cardiorespiratory fitness; Outcome: data characterizing cardiorespiratory fitness; Study design: reviews were excluded).

### 2.2. Quality Assessment

Methodological quality was assessed using the statistical review of general papers checklist [25] which we adapted for assessing protocols. All three authors independently assessed the articles. The 15 quality assessment items, scoring one point each, were evaluated and a total score determined. The quality assessment scores included the following areas: overview, purpose, literature, design, sample size reported and justified, protocol thoroughly reported, potential confounders and biases noted, outcomes reliable and valid, including statistical significance and analyses, dropouts, clinical importance and appropriate conclusions. Studies with score 10 (65%) out of 15, or above, were considered to demonstrate sufficient data quality.

## 3. Results

### 3.1. Study Selection

In total 1300 titles were found and following removal of duplicates 747 potential studies were included for eligibility screening. Based on title and abstract 643 of these were excluded. The remaining 104 full text articles were read and 62 papers excluded. Two studies presented the same data [26,27] and were merged as Brown et al., 1997. In total we retained 41 studies in this review’s qualitative synthesis (Figure 1).

### 3.2. Quality Assessment

The quality assessment of included studies examined biases and reporting accuracy with a standardized checklist administered by our review team [28]. Thirty-five articles demonstrated sufficient data quality (10/15) of 65% (Appendix A). Five of the six articles below the sufficient data quality cut-point were published 14 years or more prior to this systematic review and scored 8 (*n* = 1) and 9 (*n* = 5). We independently scored the studies meeting our 65% data quality cut off at the following levels: 10 (*n* = 9), 11 (*n* = 12), 12 (*n* = 7), 13 (*n* = 4), and 14 (*n* = 3), with average quality score 11.4. All authors clearly stated study purpose, described test protocol in detail, reported results with statistical significance and appropriate analysis methods. Almost all studies (*n* = 40) reviewed relevant background literature, reported dropouts and gave appropriate conclusions. Many authors described the sample in detail (*n* = 39), reported clinical importance (*n* = 37) and provided valid outcome measures of the aerobic test (*n* = 36). Half of the studies discussed methods for avoiding co-intervention (*n* = 20) and contamination (*n* = 19), but only a few authors justified sample size (*n* = 7), provided reliable outcome measures of the aerobic test (*n* = 6) and described the study design (*n* = 5).

### 3.3. Protocol Description

The included papers’ analyses revealed varied protocol types measuring cardiorespiratory fitness. We divided them into three groups: protocols conducted in shallow water, deep water, and others that used special equipment like underwater treadmill or bicycle. Shallow water exercise protocols were described in 13 studies, deep water exercise in 16 studies and two studies described protocols in shallow and deep water. Ten studies required special underwater equipment to conduct the protocol. Most papers did not report any pretest screening with some studies using health questionnaires or medical screenings [10,24,26,27,29,30,31,32,33,34,35,36]. Only one study [37] reported a land treadmill stress test as a pretest screening. Familarization sessions before testing were mentioned by most authors, but more specific details were provided in only seven studies [24,26,27,33,34,38,39,40]. More than half of the studies did not report any warm up. Indications for test termination were described in most studies; however, they often lacked absolute indications (e.g. the participant’s request to stop). Protocol end point mainly occurred when the participant could not produce the cadence required or arrived at volitional exhaustion. The majority of protocols (*n* = 24) were feasible for active healthy individuals (including professional athletes), eight protocols enrolled healthy participants not necessarily active, seven protocols included individuals with specific needs (patients with coronary artery disease, older adults, individuals with spinal cord injury, overweight adults and individuals with rheumatoid arthritis) and two studies did not characterize participants other than by age and sex.

#### 3.3.1. Shallow Water Exercise (SWE)

Shallow water exercise protocols are described in details in Table 1. 

In six studies [31,47,48,49,50,51] running constituted the exercise mode and the other studies employed different movement combinations of frontal kick, cross county skiing, jumping jack, rocking horse, abductor and adductor hops. For the majority of papers metronome use ensured accurate intensity levels. Water temperature ranged between 27.5 to 32 °C and water immersion levels between waist/umbilicus to shoulders. Only six studies reported warm up [29,30,37,46,48,51]. Starting intensity and increasing intensity were based upon metronome setting or rate of perceived exertion (RPE). Metronome starting intensity ranged between 80 and 90 beats per minute and increments per stage ranged between 8 and 15 beats per minute. Time on each effort stage ranged between 1 and 8 min and one study [31] reported duration based on pool length.

#### 3.3.2. Deep Water Exercise (DWE)

Deep water exercise protocols are described in details in Table 2.

Only one study recruited individuals with spinal cord injury using modified deep water running (DWR) [58]. Twelve studies reported participant tethering to maintain upright static positioning. A buoyancy belt and/or vest facilitated participant head above water positioning during testing. Water temperature ranged between 25–33 °C and water immersion level reported between the shoulder and nose. Nine studies reported conducting warm up before testing [24,33,35,40,51,53,54,58,59]. The intensity protocol was driven by a metronome, subjectively set by the participant or set by the pulley system weights. Metronome starting intensity ranged between 72–120 beats per minute and increments per stage 6–30 beats per minute. Only one study, designed to assess cardiorespiratory fitness in individuals with spinal cord injury, was set with starting intensity at 40 beats per minute [58]. Time on each stage of the protocol ranged between 1–4 min.

#### 3.3.3. Other Protocols

Protocols with the use of special equipment are described in details in Table 3.

The protocols with special equipment used underwater treadmill [32,38,60,61,65,66] and underwater bicycle [62,63,64,67] with water temperature of 28–30 °C, with one study [38] ranging between 20.6–35.6 °C. Water immersion level was reported at the xiphoid process, and intensity set by treadmill speed or cycling cadence. In two studies [38,60] additional water jet resistance was used. Five studies reported warm up [38,61,63,65,66]. Costa et al. [63] conducted three protocols with different frontal surface areas while cycling. Time on each exercise stage ranged between 1–3 min.

### 3.4. Peak Outcomes

Authors of 37 studies provided mean VO_2peak_/VO_2max_ values (Table 4) and other test outcomes considered as secondary criteria used when an oxygen uptake plateau was not evident. Varying outcomes in VO_2peak_/VO_2max_ were reported in the included studies.

For 12 protocols based on SWE (*n* = 8) and SWR (*n* = 4), mean VO_2peak_/VO_2max_ ranged from 20.6–57.2 mL/kg/min for young (often trained) participants (19–26 years old) and 21.8–45.94 mL/kg/min for older participants (40–66 years old, including competitive runners). Other SWE and SWR outcome measures were HRmax in 11 studies (range 156–192 bpm), mean RER in 8 studies (range 1–1.38), mean time to exertion in two studies (range 8.52−9.93 min) and RPE in eight studies (range 9.6–19.7). Six studies compared SWE and SWR protocol with land protocol and indicated lower aquatic environment VO_2peak_/VO_2max_ values [31,42,43,47,48,50].

DWE (*n* = 1) and DWR (*n* = 16) reported VO_2peak_/VO_2max_ range 20.32–48.8 mL/kg/min. Some studies reported VO_2peak_/VO_2max_ in L/min, with values from 1.3–4.03 L/min. Other researchers who utilized outcome measures for DWE and DWR protocols stated HRmax in 14 studies (range 135–183.4 bpm), mean RER in 9 studies (range 0.93–1.28), mean time to exertion in two studies (range 4.78–13.83 min) and RPE in six studies (range 9.5−19.3). Sixteen studies compared DWR and DWE protocol with land protocol and indicated lower aquatic environment VO_2peak_/VO_2max_ values.

In 9 of 10 protocols including underwater treadmill running (UTM) (*n* = 5) and water cycling (*n* = 4) VO_2peak_/VO_2max_ ranged from 28.64–62.2 mL/kg/min. Other UTM and water cycling protocol outcome measures encompassed mean HRmax in nine studies (range 131.9–188.8 bpm), mean RER in five studies (range 0.97–1.15), mean time to exertion in four studies (range 8.8−14.87 min) and RPE in eight studies (range 17–19). Five studies compared UTM and water cycling protocol with land protocol and indicated similar VO_2peak_/VO_2max_ values.

Twenty-eight of 37 studies predefined the criteria for reaching a valid VO_2peak_/VO_2max_. Varying criteria included: RER above a certain level (>1.0–1.15) (*n* = 20), reaching a VO_2_ plateau (*n* = 18), attainment of age-predicted maximal heart rate (*n* = 9), RPE above level of 17–18 in Borg’s 6–20 RPE Scale (*n* = 7), highest observed VO_2_ value measured (*n* = 6), maximal respiratory rate of at least 35 breaths per minute (*n* = 4) and blood lactate level above 8–9 mmol/l (*n* = 2). Other criteria similar to the previously described test termination criteria, included general exhaustion [24,33], RPE equals 10 in Borg’s 0–10 RPE Scale [63] or inability to maintain the required cadence or load [41,52,55]. One study referred to the American College of Sports Medicine (ACSM) guidelines [61]. The number of required criteria for reaching a valid VO_2peak_/VO_2max_ ranged from one specific criterion (seven studies), through 2–4 criteria (11 studies), to selected 1–3 criteria from the list of 3–5 criteria (nine studies).

## 4. Discussion

This systematic literature review provided an overview of the published head-out water-based protocols used to assess cardiorespiratory fitness. The results indicated varied head-out protocols were used in the aquatic setting to assess cardiorespiratory fitness. The majority of tests were conducted in a non-laboratory setting, using clinical or performance exercise incremental step test protocols for an aquatic environment. Based on our analyses we provide the following suggestions for testing cardiorespiratory fitness in water, which may help researchers and clinicians.

The physiological responses to head-out water-based exercises are temperature dependent [18]. In the analyzed papers the water temperature ranged between 25 and 33 °C. The thermoneutral temperature during exercise is considered between 28 and 30 °C. The temperatures below and above this level additionally impact the physiological response, particularly vasoconstriction and vasodilation. Maintaing the same water temperature for all the participants is also crucial. Most analyzed protocols were conducted in the water with a temperature difference of 1 °C. One study [38] reported water temperature difference of 15 °C for participants. Secondary to water temperature’s influence upon physiological responses, large temperature differences should be avoided during testing protocols. The recommended water temperature for cardiorespiratory fitness testing is 28–30 °C with difference of maximum 1 °C between testing participants, which the analyzed studies mainly incorporated.

Another physiological response modulator was water depth and resultant hydrostatic pressure [18]. The biggest water depth differences were found in the shallow water protocols. The water level was set between waist/umbilicus to shoulders. Oxygen uptake and energy expenditure are lower at breast immersion when compared with hip immersion [68]. Heart rate decreases significantly with the increase of body immersion [69] with physiological demand apparently lower for deep-water versus shallow-water exercises [70]. Only one study compared these two modes of exercise and provided cardiorespiratory fitness values [47] confirming higher values for shallow water running. Deep water protocols are better suited for participants who are already familiar with exercising in deep water or professional running, in the case of DWR. In our review, the highest values of mean reported VO_2peak_/VO_2max_ achieved in DWR protocols were observed for individuals trained in water running [39], active males [54] and male runners [59]. On the other hand, deep water testing is a choice for people who are unable to perform high intensity cardiorespiratory movement in shallow water, such as individuals with spinal cord injury [58].

Interestingly, when the pattern of movement and protocols accurately matched water, and on land, no differences in VO_2peak_/VO_2max_ values occurred between conditions [18]. This aligns with the present review findings of underwater and land treadmills [32,38,61,66], with stationary water running and stationary running on land [50] and with water cycling and bicycle ergometer [67]. Therefore, a determining factor for the VO_2peak_/VO_2max_ pattern is the mode of exercise performed rather than inherent water properties.

In our review, mean reported VO_2peak_/VO_2max_ values achieved in SWE protocols differed when varied movements were performed [41,42,43,45]. For these studies we reviewed exercise type to understand the 10–15% lower VO_2peak_/VO_2max_ achieved. No patterns arose except with varied exercise, it may be more difficult to maintain exercise intensity stage even with metronome guidance. It seemed wider movements like jumping jacks or abductor/adductor hops might be more difficult in water with higher intensity due to water resistance. Running movement is natural for humans and based on the critically assessed data we suggest that cardiorespiratory fitness testing in shallow water incorporates running. However, with this limited data no final conclusions can be drawn.

The analysis of criteria used for achieving VO_2peak_/VO_2max_ during head-out water-based exercise protocols indicated lack of uniformity for both plateau definition and secondary criteria for reaching a valid VO_2peak_/VO_2max_. Similar to the land environment it is common in water for participants to complete a maximal graded exercise test and fail to achieve a plateau in VO_2_ [71]. Thus, secondary criteria variety for reaching a valid VO_2peak_/VO_2max_ exists in head-out water-based protocols. However, no general agreement on specific secondary criteria alone, or in combination, arose; moreover, criteria selection appeared arbitrary without justification and/or scientific evidence. The terminology associated with measuring maximal oxygen uptake in water environment was also inconsistent. The interchangeable use of terms VO_2max_ and VO_2peak_ neglects original definitions and adds confusion in the literature.

Correctly setting the intensity and increments per testing stage is one of the factors affecting cardiorespiratory fitness values. Most studies used a metronome as an objective tool to conduct the test, using small to modest individualized increments per stage, resulting in test completion between 8 and 12 min [2], the increments per stage ranged between 6 and 30 beats per minute. The time until exhaustion, in the study which used 30 beats per minute increment, was 7.3 min [24], which according to the guidelines might not be enough to reach VO_2peak_ individual values. Based on this literature review, use of 10–15 beats per minute increments while using the metronome is optimal. The starting intensity requires individual adjustment to the population tested.

Appraising study quality is difficult but necessary. We provided quality scores but did not create a high or low binary categorization [72]. The three consistent quality areas of concern included: study design type, sample size justification and reliability of outcome measures. Future researchers and clinicians need to carefully analyze all data to determine which head-out of water cardiorespiratory protocol best fits their needs.

Generally, in studies analyzing cardiorespiratory fitness protocols used for land exercise, testing does not always comply with exercise testing guidelines [7,71] and parallels our findings as only two studies [32,61] followed the commonly accepted ACSM exercise testing guidelines. To implement testing protocols in the aquatic environment the following standard steps are required: pretest screening to identify contraindications for maximal exercise, familiarization session, indications for test termination, warm up, load increments per stage (resulting in completion of the test between 8 and 12 min), and criteria used to verify achievement of VO_2peak_/VO_2max_. Additionally, these testing components require modification according to the individual exercise test purpose and participants’ ability.

### Limitations

This review included only articles in English found in five databases. It is possible that some relevant studies were not included in the search strategy used. However, with the detailed search terms used, the screening performed by two independent reviewers, and only three articles excluded due to language, the risk of selection bias was limited. The adapted methodological quality assessment with an arbitrarily set data quality cut point potentially was influenced by reporting and interpretation bias. The selected studies’ heterogeneity included differences in study protocols, outcome measures, and statistical and analytical methods which limited advanced comprehensive statistical analyses and interpretation.

## 5. Conclusions

Analyzed protocols were highly diverse and no one broadly accepted head-out water-based protocol exists to evaluate cardiorespiratory fitness and compare results. However, three groups were differentiated: protocols conducted in shallow water, deep water, and others that used special equipment. Moreover, based on analyzed protocols, three key testing proprieties for cardiorespiratory fitness testing can be suggested for clinical use: water temperature of 28–30 °C with difference of maximum 1 °C between testing participants and/or testing sessions, water depth adapted for participant aquatic experiences and abilities, and intensity of 10–15 metronome beats per minute increment. The available data supported tat RPE, HR, and VO_2peak_/VO_2max_ should all be included in aquatic environment cardiorespiratory fitness testing, similarly to land testing. Future research is needed to test methodological standardization in which protocols can be individualized to specific populations, and to determine reliability and validity of the specific protocols. Furthermore, consensus regarding reporting test procedures, outcomes and proprieties in terms of gold standard is necessary to enhance comparison and understanding of intervention results.

## Figures and Tables

**Figure 1 ijerph-17-07215-f001:**
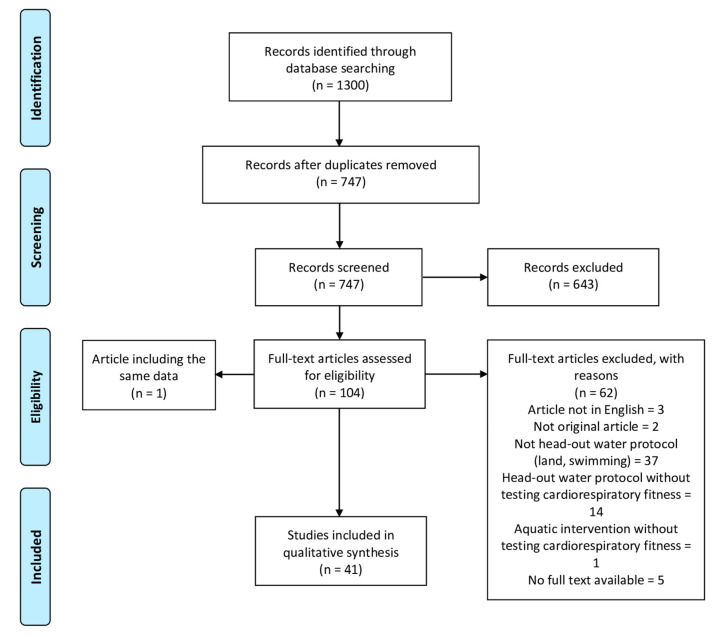
Study selection flow diagram.

**Table 1 ijerph-17-07215-t001:** Test protocols in shallow water.

Author, Year	Mode of Exercise	Pretest Screening	Familiarization	Equipment	Water Temperature [°C]	Water Depth [m]	Water Immersion Level	Warm Up	Starting Intensity	Time on Starting Intensity	Increasing Intensity	Time on Each Next Stage	Test Termination Criteria
Alberton et al., 2013 [41]	stationary running (SR) frontal kick (FK) cross country skiing (CCS)	NR	yes	metronome	31–32	0.95–1.30	xiphoid process	NR	80 cpm	2 min	10 cpm	1 min	exhaustion
Alberton et al., 2013 [42]	stationary running (SR) frontal kick (FK) jumping jack (JJ)	NR	yes	metronome	31–32	0.95–1.30	between the xiphoid process (while standing) and the shoulders (while exercising)	NR	85 bpm	3 min	15 bpm	2 min	exhaustion or unable to keep up with the cadence
Alberton et al., 2014 [43]; Alberton et al., 2016 [44]	stationary running (SR) frontal kick (FK) cross country skiing (CCS)	NR	yes	metronome	32	0.95–1.30	xiphoid process	NR	80 bpm	2 min	10 bpm	1 min	exhaustion or unable to keep up with the cadence
Antunes et al., 2015 [45]	stationary running (SR) frontal kick (FK) cross country skiing (CCS) adductor hop (ADH) abductor hop (ABH) jumping jacks (JJ)	NR	yes	metronome	31–32	0.95–1.30	between the xiphoid process (while standing) and the shoulders (while exercising)	NR	80 bpm	2 min	10 bpm	1 min	exhaustion or unable to keep up with the cadence
Bartolomeu et al., 2017 [46]	Rocking horse	NR	NR	metronome	31	NR	xiphoid process	6 min at starting intensity	90 bpm	6 min	15 bpm	6 min	unable to keep up with the cadence for more than 30 s
Campbell et al., 2003 [37]	Walking/running with different arms movement	medication cardiac, PARQ; older group: modified Bruce treadmill stress test	NR	webbed gloves for maximal shallow water run	27.5–28.0	NR	xiphoid process to axilla	5–10 min	self-selected; carry on conversation	8 min	5 different submax activities; 10−15 min rest; one maximal 300 m shallow jogging	8 min	NR
D’Acquisto et al., 2015 [29]	Bout 1: Jog Bout 2: Tuck Jumps With Plunge Bout 3: Cross-Country Ski Bout 4: Deep Split Jump Lunge Bout 5: Alternating Long Leg kicks	health history, physical activity questionnaire	yes	NR	28.6 (0.3)	NR	axilla	instructor guided 6 min	RPE 9	5 min	RPE 11, 13, 15, 17	5 min	NR
Dowzer et al., 1999 [47]	SWR	NR	yes	water shoes, metronome	29	1.2	waist	NR	132 strides/min	1 min	12 strides/min 8 strides/min	1 min	volitional exhaustion
Fisher et al., 2019 [30]	RPE 9: Jog with swinging arms RPE 11: Tuck Jumps with Plunge RPE 13: Cross-Country Ski RPE 15: Deep Jump Lunge RPE 17: Alternating Long Leg Kicks	health history, physical activity questionnaire	yes	webbed gloves	28.3–28.9	NR	axilla	6 min	RPE 9	5 min	RPE 9, 11, 13, 15, 17, 20	5 min, 1 min rest between	NR
Conti et al., 2008 [48]	SWR	NR	yes	metronome	29–30	1.2	lower than xiphoid process	3 min self-selected stride frequency	12 strides additional from warm-up	1 min	12 strides/min	1 min	NR
Conti et al., 2015 [49]	SWR	NR	yes	metronome	29.5 (2.0)	NR	umbilicus	NR	NR	5 min	NR	5 min	NR
Kruel et al., 2013 [50]	SWR	NR	NR	NR	31–32	NR	xiphoid process	NR	85 bpm	2 min	15 bpm	2 min	exhaustion
Nagle et al., 2017 [31]	SWR	medical inventory and PARQ	yes	water exercise shoes	27.5	1.2	below xiphoid process to the midaxillary region	NR	4 in OMNI scale 1–10 RPE	4 × 22 m	stage 1: 4 lengths, 10 s rest stage 2: 3 lengths, 5 s rest stage 3: 2 lengths, 3−5 s rest stage 4: 4−6 continuous	stage 1: 4–6 OMNI stage 2: 6–8 OMNI stage 3: 8–9 OMNI stage 4: > 9 on the OMNI	volitional fatigue
Town et al., 1991 [51]	SWR	NR	yes	NR	NR	1.3	NR	6 min	NR	NR	subjectively	1 min (final stage—2 min)	NR

ABH-abductor hop, ADH-adductor hop, bpm-beats per minute, CCS-cross country skiing, cpm-cycles per minute, FK-frontal kick, JJ-jumping jack, NR-not reported, PARQ-Physical Activity Readiness Questionnaire, RPE-rating of perceived exertion, SR-stationary running, SWR-shallow water running.

**Table 2 ijerph-17-07215-t002:** Test protocols in deep water.

Author, Year	Mode of Exercise	Pretest Screening	Familiarization	Equipment	Water Temperature [°C]	Water Depth [m]	Water Immersion Level	Warm Up	Starting Intensity	Time on Starting Intensity	Increasing Intensity	Time on Each Next Stage	Test Termination Criteria
Broman et al., 2006 [33]	DWR *	medical screening, exercise history questionnaire	2 (25 min and 30 min)	vest, elastic cord	27	NR	NR	10 min	stride frequency at the first stage of individual values of oxygen uptake measured on the treadmill	4 min	increasing the stride frequency to the stage of the submaximal individual values of oxygen uptake measured on the treadmill	4 min (1 min rest between each stage)	inability to continue running
Brown et al., 1997 [26,27]	DWR *	health history questionnaire	2 × 30 min	Aqua Jogger, belt, metronome	29.6 (0.5)	NR	NR	NR	72 spm	3 min	12 spm	3 min	unable to keep proper cadence or DWR form
Brown et al., 1998 [52]	DWR	NR	yes	Aqua Jogger, belt, metronome	27.9–28.5	NR	NR	NR	72 spm	3 min	12 spm	3 min	unable to keep proper cadence or DWR form, VO_2_ plateau
Butts et al., 1991 [53]; Butts et al., 1991 [54]	DWR *	yes	yes	Wet Vest, rope	29	NR	NR	5 min at 100 bpm	120 bpm	2 min	20 bpm	2 min	request of the subject or objective signs of exhaustion
Dowzer et al., 1999 [47]	DWR *	NR	yes	wet vest, rope, metronome	29	NR	chin and nose level	NR	120 strides/min	1 min	12 strides/min 8 strides/min	1 min	volitional exhaustion
Frangolias et al., 1996 [39]	DWR	water running style	yes, at least 3 sessions	buoyancy belt, sponges to maintain fists	28	NR	head above water	NR	NR	NR	NR	NR	NR
Gayda et al., 2010 [24]	3 DWR * protocols (short, intermediate, long)	medical screening	yes (5 min before first test)	metronome, floatation vest–wet, elastic cord	30	NR	neck	2 min	56 cpm	2 min	8–30 cpm dependent upon protocol (short, intermediate, long)	2 min	voluntary signal participant and/or unable to maintain cadence
Kanitz et al., 2015 [55]	DWR	NR	yes	float vest, cable, metronome	30	2	shoulder	NR	85 bpm	3 min	15 bpm	2 min	subject reached the maximum effort
Melton-Rogers et al., 1996 [56]	DWR *	NR	NR	wet vest, elastic cord	33	NR	neck	NR	92 bpm	2 min	6 bpm	2 min	NR
Mercer et al., 1998 [57]	DWR *	NR	yes	Aqua Jogger belt, bucket, wooden plank	26.9	NR	NR	NR	0.57 kg weight	1 min	0.57 kg weight	1 min	unable to keep the bucket from touching the deck
Meredith-Jones et al., 2009 [36]	DWR *	modified PARQ	yes	bucket, wooden plank	29	1.8	neck level C7	NR	0.57 kg weight	1 min	0.57 kg weight	1 min	unable to keep the bucket from touching the wooden plank
Michaud et al., 1995 [40]	DWR *	NR	yes (3 sessions)	wet vest, bucket, tether, pulleys	29–30	3.66	head above water	yes	individually based on familiarization session	individually based on familiarization session	individually based on familiarization session	3 min	unable to maintain proper running form, unable to remain in target area
Michaud et al., 1995 [34]	DWR *	clearance by physician	2–3 × 20−30 min	Aqua Jogger, metronome, headphones, testing frame	27–29	NR	head above water	NR	48 cpm	3 min	66, 72, 76, 80, 84 cpm	3 min	signal by the subject
Ogonowska-Slodownik et al., 2019 [58]	DWE (arms, trunk, legs) *	NR	yes	foam dumbbells, weights, flotation belt, metronome	31–32	2.13	NR	3 min	40 bpm	3 min	10 bpm	1 min	volitional fatigue or unable to perform the required work rate
Phillips et al., 2008 [35]	DWR *	medical screening, exercise history questionnaire	yes	flotation belt, bucket, wooden plank	29	1.8	NR	individual	0.57 kg weight	1 min	0.57 kg weight	1 min	unable to keep the bucket from touching the wooden plank
Svedenhag et al., 1992 [59]	DWR	NR	yes	buoyancy jacket	25	NR	NR	5 min	115 bpm	4 min	stage 1: 115 bpm, 1 min rest; stage 2: 130 bpm, 1 min rest; stage 3: 145 bpm, 1 min rest; stage 4: 155 bpm, 3–4 min rest; stage 5: maximal intensity; stage 6: exhaustion	4 min + 1–2 min + 1 min	exhaustion
Town et al., 1991 [51]	DWR	NR	yes	NR	NR	2.5–4	NR	6 min	NR	NR	subjectively every minute	1 min (final stage—2 min)	NR

*-tethered, bpm-beats per minute, cpm-cycles per minute, DWE-deep water exercise, DWR-deep water running, NR-not reported, PARQ-Physical Activity Readiness Questionnaire, RPE-rating of perceived exertion, spm-steps per minute.

**Table 3 ijerph-17-07215-t003:** Test protocols with special equipment.

Author, Year	Mode of Exercise	Pretest Screening	Familiarization	Equipment	Water Temperature [°C]	Water Depth [m]	Water Immersion Level	Warm Up	Starting Intensity	Time on Starting Intensity	Increasing Intensity	Time on Each Next Stage	Test Termination Criteria
Brubaker et al., 2011 [60]	UTM	NR	NR	underwater treadmill (Hydro-worx 1000), resistance jets	28	NR	xiphoid process	NR	2.3 km/h	2 min	2.3 km/h 4.9 km/h 7.3 km/h 9.6 km/h 9.6 km/h, 30% resistance jets 9.6 km/h, 40% resistance jets 9.6 km/h, 50% resistance jets	2 min	NR
Choi et al., 2015 [61]	UTM	NR	NR	underwater treadmill (Focus, Hydro-physio)	28	NR	midpoint umbilicus and xiphoid process	5 min	2.0 km/h	1 min	0.5 km/h	1 min	participant request and/or ACSM guidelines
Colado et al., 2019 [62]	water cycling	NR	yes	underwater bicycle (Hydro-rider), metronome	30	NR	xiphoid process	NR	100 bpm	3 min	15 bpm	2 min	exhaustion, not maintaining pedal rate
Costa et al., 2017 [63]	water cycling Frontal surface area (FSA) FSA1: 500cm2 FSA2: 580cm2 FSA3: 660cm2	NR	NR	underwater bicycle (Hydro-cycle)	28	NR	xiphoid process	5 min/50 rpm	50 rpm	1 min	3 rpm	1 min	exhaustion
Greene et al., 2011 [32]	UTM	stratified according to ACSM standards for risk of cardiovascular disease, physiological examination	yes	underwater treadmill (Hydro-worx 1000 and 2000)	32–34	NR	fourth intercostal space	NR	53.6 m·min-1	3 min	26.8 m·min−1	3 min	voluntary exhaustion, the exercise protocol completed
Pinto et al., 2016 [64]	water cycling	NR	yes	underwater bicycle (Hydro-rider), compact disc	30	NR	NR	NR	100 bpm	3 min	15 bpm	2 min	exhaustion
Schaal et al., 2012 [38]	UTM (with and without underwater running shoes)	NR	2 × 5 min	underwater treadmill (Hydro-worx 1000), water running shoes	20.6–35.6	NR	xiphoid process	1–5 min	0.5 mph and 40% water jets	1 min	0.5 mph every min for 4−5 min, then 0.5 mph every min and 10% water jets every min	1 min	volitional or treadmill’s speed 7.5 mph reached and maintained for minute
Silvers et al., 2007 [65]; Silvers et al., 2008 [66]	UTM	NR	NR	underwater treadmill (Hydro-worx 2000), resistance jets	28	NR	xiphoid process	6 min	13.4 m/min and 40% water jets	1 min	13.4 m/min every min for 4−5 min, then 13.4 m/min every min and 10% water jets every min	1 min	volitional exhaustion
Yazigi et al., 2013 [67]	water cycling (at two water temperature)	NR	yes	underwater bicycle (Hydro-rider)	27 and 31	adapted	xiphoid process	NR	50 rpm	3 min	10 rpm till 70 rpm; 5 rpm after 70 rpm	3 min	unable to maintain the cadence

ACSM—American College of Sports Medicine, bpm-beats per minute, FSA- Frontal Surface Area, mph—miles per hour, NR-not reported, rpm-repetitions per minute, UTM-underwater treadmill running.

**Table 4 ijerph-17-07215-t004:** Population characteristics and cardiorespiratory fitness values.

Author, Year	Mode of Exercise	Population		AQUA					LAND
		Characteristic	Age of Participants Mean (SD)	Mean VO_2peak_/VO_2max_ (SD) (mL·kg^–1^·min^–1^)	Mean HRmax (SD) (bpm)	Mean RER (SD)	Mean Time to Exhaustion (SD) (min)	Mean RPE (SD)	Mean VO_2peak_/VO_2max_ (SD) (mL·kg^−1^·min^−1^)
Alberton et al., 2013 [41]	SWE	F = 20 (active)	24 (2.5)	SR: 30.31 (5.21) FK: 30.95 (3.61) CCS: 29.88 (4.44)	SR: 186 (7) FK: 184 (7) CCS: 182 (12)	SR: 1.38 (0.11) FK: 1.32 (0.09) CCS: 1.27 (0.12)	SR: 9.93 (1.59) FK: 8.52 (1.23) CCS: 8.58 (1.21)	SR: 18.85 (0.49) FK: 18.80 (0.52) CCS: 18.75 (0.55)	NA
Alberton et al., 2013 [42]	SWE/LTM	F = 9	22.89 (1.81)	SR: 34.00 (3.90) FK: 33.77 (2.74) JJ: 23.95 (3.09)	NR	NR	NR	SR: 19.22 (0.42) FK: 18.67 (0.48) JJ: 18.89 (0.32)	39.32 (3.70)
Alberton et al., 2014 [43]	SWE/LTM	F = 20 (active)	24 (2.5)	SR: 30.31 (5.21) FK: 30.95 (3.61) CCS: 29.88 (4.44)	SR: 185.94 (6.99) FK: 184.31 (7.16) CCS: 182.25 (12.21)	SR: 1.38 (0.11) FK: 1.32 (0.09) CCS: 1.27 (0.12)	SR: 9.93 (1.59) FK: 8.52 (1.23) CCS: 8.58 (1.21)	SR: 18.85 (0.49) FK: 18.80 (0.52) CCS: 18.75 (0.55)	36.03 (4.10)
Antunes et al., 2015 [45]	SWE	F = 12 (active, students)	24 (2)	SR: 28.9 (4.7) FK: 30.2 (2.5) CCS: 29.1 (3.3) ADH: 24.5 (6.3) ABH: 25.2 (3.6) JJ: 20.6 (4.1)	SR: 186.3 (5.0) FK: 184.6 (5.1) CCS: 183.8 (8.8) ADH: 180.7 (9.3) ABH: 178.5 (9.8) JJ: 167.8 (16.2)	NR	NR	NR	NA
Bartolomeu et al., 2017 [46]	SWE	F young = 19 F older = 18	young: 22.16 (2.63) older: 65.06 (5.77)	young: 44.49 (1.88) older: 32.98 (1.72)	young: 192.49 (1.89) older: 162.46 (4.14)	NR	NR	young: 16.42 (1.61) older: 15.67 (1.53)	NA
Broman et al., 2006 [33]	DWR/LTM	F = 11 (older women)	70 (2)	1.30 (0.14) L/min	NR	NR	NR	NR	1.82 (0.2) L/min
Brown et al., 1997 [26,27]	DWR/LTM	F = 12 M = 12 (healthy, recreational exercisers)	F: 21 (1.9) M: 20 (0.8)	F: 30.1 (4.5) M: 39.1 (8.3)	F: 173.9 (7.3) M: 183.8 (7.7)	NR	NR	NR	F: 40.1 (3.1) M: 45.2 (4.4)
Brown et al., 1998 [52]	DWR	F = 15 M = 18 (students)	F: 21.3 (2.2) M: 22.8 (2.7)	F: 34.1 (5.0) M: 37.1 (7.1)	F: 181 (8) M: 176 (9)	NR	NR	NR	NA
Butts et al., 1991 [53]	DWR/LTM	F = 12 (runners)	15.4 (1.1)	48.8 (9.1)	180.3 (8.0)	1.01 (0.08)	NR	19.3 (0.6)	54.7 (7.0)
Butts et al., 1991 [54]	DWR/LTM	F = 12 M = 12 (active)	F: 21.9 (2.4) M: 20.6 (1.9)	F: 46.8 (5.9) M: 58.4 (3.9)	F: 179 (7.5) M: 183.4 (5.9)	F: 1.09 (0.04) M: 1.11 (0.03)	NR	NR	F: 55.7 (4.8) M: 64.5 (2.8)
Campbell et al., 2003 [37]	SWE	F young = 11 F older = 11	young: 21.3 (0.4) older: 66.7 (0.9)	young: 37.90 (2.21) older: 21.80 (0.91)	young: 181.7 (2.7) older: 156.2 (4.5)	young: 1.11 (0.06) older:1.19 (0.04)	NR	young: 17.0 (0.4) older: 17.2 (0.6)	NA
Choi et al., 2015 [61]	UTM/LTM	F = 4 M = 17 (patients with coronary artery disease)	59.9 (9.1)	29.8 (4.8)	131.9 (13.7)	0.97 (0.07)	12.4 (3.7)	17.0 (0.9)	31.1 (5.3)
Colado et al., 2019 [62]	Water cycling	M = 30 (active students)	22.37 (2.31)	46.89 (5.64)	173 (28)	NR	NR	19.0 (0.71)	NA
Conti et al., 2008 [48]	SWR/LTM	M = 12 untrained (UT) and trained (T) (pentathlon Olympic athletes)	UT: 22 (1.0) T: 19 (1.0)	UT: 45.2 (6.8) T: 57.2 (3.9)	UT: 182 (8.9) T: 177 (7.1)	UT: 1.1 (0.09) T: 1.0 (0.04)	NR	NR	UT: 47.9 (3.6) T: 68.9 (5.1)
Costa et al., 2017 [63]	Water cycling	M = 15 (healthy)	24.1 (4.0)	FSA1: 44.2 (7.3) FSA2: 45.0 (7.8) FSA3: 44.2 (6.6)	FSA1: 182 (10) FSA2: 183 (8) FSA3: 183 (10)	NR	NR	FSA1: 9.8 (0.4) FSA2: 9.9 (0.3) FSA3: 9.9 (0.3)	NA
D’Acquisto et al., 2015 [29]	SWE	F = 9 (healthy, active)	26 (6)	41.3 (4.6)	181 (7)	1.05 (0.05)	NR	19.7 (0.5)	NA
Dowzer et al., 1999 [47]	SWR, DWR/LTM	M = 15 (competitive runners)	40.93 (9.48)	SWR: 45.94 (6.1) DWR: 41.27 (6.4)	SWR: 165 (16) DWR: 153 (16)	SWR: 1.07 (0.1) DWR: 1.08 (0.1)	NR	NR	55.39 (8.46)
Fisher et al., 2019 [30]	SWE	F = 9 M = 9 (active)	F: 26 (6) M: 24 (1)	F: 41.3 (4.6) M: 42.8 (4.7)	F: 181 (7) M: 185 (7)	F: 1.05 (0.05) M: 1.08 (0.06)	NR	F: 19.7 (0.5) M: 19.4 (0.5)	NA
Frangolias et al., 1996 [39]	DWR/LTM	F = 8 M = 14 UT (untrained in DWR) = 6 T (trained in DWR) = 16	UT: 26.3 (4.7) T: 26.7 (4.7)	UT:53.5 (6.2) T: 53.8 (5.4)	UT: 173.8 (10.1) T: 172.6 (14.0)	UT:1.14 (0.04) T: 1.12 (0.04)	NR	NR	UT: 63.8 (3.0) T: 58.8 (6.2)
Gayda et al., 2010 [24]	DWR/LTM	F = 13 M = 11 (healthy, active, older)	60 (6)	S: 27.83 (8.03) I: 26.22 (7.04) L: 26.33 (8.1) (ml/LBM/min)	S: 139 (14) I: 141 (11) L: 135 (16)	S: 0.93 (0.09) I: 0.99 (0.16) L: 1.00 (0.14)	S: 7.3 I: 10.31 L: 13.86	NR	49.39 (14.4) (ml/LBM/min)
Greene et al., 2011 [32]	UTM/LTM	F = 28 M = 27 (healthy)	F: 41 (12) M: 41 (14)	F: 28.64 (9.0) M: 33.32 (7.35)	F: 167 (17) M: 167 (16)	F: 1.02 (0.09) M: 1.05 (0.08)	NR	F: 17 (2) M: 17 (2)	F: 27.55 (7.90) M: 33.02 (8.63)
Kanitz et al., 2015 [55]	DWR/LTM	F = 12 (active)	23.2 (1.9)	22.5 (4.1)	174 (9)	NR	NR	NR	33.7 (3.9)
Kruel et al., 2013 [50]	SWR/LTM, SRL	F = 9 (active)	23 (1.9)	34.00 (3.9)	187.25 (6.75)	NR	NR	NR	LTM: 38.98 (3.39) SRL: 34.88 (3.64)
Melton-Rogers et al., 1996 [56]	DWR/bicycle ergometry	F = 8 (rheumatoid arthritis)	35.88 (2.85)	20.32 (7.33)	178.75 (30.10)	1.28 (0.27)	NR	18.13 (1.73)	23.35 (9.17)
Mercer et al., 1998 [57]	DWR/LTM	F = 13 M = 15	F: 21 (1.3) M: 24.3 (4.7)	44 (10)	177 (9)	NR	NR	NR	54 (13)
Meredith-Jones et al., 2009 [36]	DWR	F = 18 (sedentary, overweight)	59 (8.6)	*pre: 1.37 (0.10) L.min-1 *post: 1.51 (0.08) L.min−1	NR	NR	NR	NR	NA
Michaud et al., 1995 [40]	DWR/LTM	M = 6 (runners)	25.5 (5.1)	3.8 (0.11) L/m	168.8 (5.2)	1.0 (0.08)	NR	9 (1)	4.3 (0.10) L/m
Michaud et al., 1995 [34]	DWR/LTM	F = 8 M = 2 (healthy, sedentary)	32.6 (6.8)	*pre: 1.79 (0.59) L.min−1 *post: 2.15 (0.59) L.min−1	*pre: 172 (16.7) *post: 175 (13.9)	*pre: 1.21 (0.12) *post: 1.24 (0.09)	NR	*pre: 9.5 (0.85) *post: 9.9 (0.31)	*pre: 2.25 (0.57) L.min−1 *post: 2.49 (0.68) L.min−1
Nagle et al., 2017 [31]	SWR/LTM	F = 23 (healthy)	20 (3)	37.1 (6.8)	181 (11)	1.09 (0.12)	NR	9.6 (0.8)	44.2 (8.4)
Ogonowska-Slodownik et al., 2019 [58]	DWE/arm cycle ergometry	F = 4 M = 13 (spinal cord injury)	45.7 (11.6)	test 1: 18.63 (5.26) test 2: 18.17 (5.49)	NR	NR	NR	17.41 (2.88)	test 1: 1.54 (5.00) test 2: 17.68 (5.55)
Phillips et al., 2008 [35]	DWR/LTM	F = 20 (healthy, overweight)	48.0 (7.1)	22.5 (4.9)	159 (16)	1.03 (0.06)	4.78 (1.32)	17 (2)	27.7 (4.7)
Pinto et al., 2016 [64]	Water cycling	M = 27 (fit university students)	22.46 (2.35)	55.04 (8.64)	186 (10)	NR	14.87 (1.75)	NR	NA
Schaal et al., 2012 [38]	UTM/LTM	M = 14 (triathletes)	35.1 (9.8)	without: 51.77 (8.7) with: 53.2 (6.8)	without: 172.8 (9.7) with: 172.5 (12.8)	without: 1.12 (0.09) with: 1.13 (0.10	NR	without: 18.5 (1.1) with: 19.3 (0.77)	53.01 (6.9)
Silvers et al., 2007 [65]	UTM/LTM	F = 11 M = 12 (recreationally competitive runners)	F: 22.1 (2.3) M: 24.8 (3.8)	52.8 (7.7)	188.8 (10.4)	1.15 (0.04)	8.8 (1.5)	18.4 (1.4)	52.5 (8.4)
Silvers et al., 2008 [66]	UTM	F = 11 M = 13 (recreationally competitive runners)	25 (3)	trial 1: 3.65 (0.80) L/min trial 2: 3.67 (0.80) L/min	trial 1: 187 (13) trial 2: 187 (14)	trial 1: 1.12 (0.05) trial 2: 1.12 (0.10)	trial 1: 10.0 (1.3) trial 2: 10.2 (1.3)	trial 1: 19 (1) trial 2: 19 (1)	NA
Svedenhag et al., 1992 [59]	DWR/LTM	M = 10 (runners)	26.4	4.03 (0.13) l/min	172 (3)	NR	NR	NR	4.60 (0.14)
Yazigi et al., 2013 [67]	Water cycling/bicycle ergometry	M = 10 (active students)	22 (1)	27 °C: 52.5 (10.1) 31 °C: 62.2 (12.4)	27 °C: 188 (13) 31 °C: 185 (9)	NR	NR	NR	62.2 (10.1)

ABH-abductor hop, ADH-adductor hop, CCS-cross country skiing, DWE-deep water exercise, DWR-deep water running, F-female, FK-frontal kick, FSA-frontal surface area, HRmax-Heart Rate maximum, JJ-jumping jack, LTM-land treadmill running, M-male, NA-not analyzed, NR-not reported, RPE-rating of perceived exertion, SD-standard deviation, SR-stationary running, SRL-stationary running on land, SWE-shallow water exercise, SWR-shallow water running, T-trained, UT-untrained, UTM-underwater treadmill running, VO_2peak_/VO_2max_-peak/maximal oxygen consumption.

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
