# Peer review of "Head-Out Water-Based Protocols to Assess Cardiorespiratory Fitness—Systematic Review"

_ijerph, 2020, doi:10.3390/ijerph17197215_

Round 1

Reviewer 1 Report

This review investigated the different head-out water-based protocols used to assess cardiorespiratory fitness. I feel that this is an interesting paper about the possibility to use aquatic environment to perform tests and training for different populations. The study presented an overall experienced large number of papers with high quality and criteria. Also, there is no review about the theme. Therefore, while the manuscript was generally well written and fill a gap in the scientific literature, I believe that there are currently a number of issues that need to be addressed. Major concerns are also provided in a number of specific comments below.

Normally, the introduction has 4-5 paragraphs where the authors try to explain for the readers the background about theme. Please, provide more evidences.

It also needs to be clear what the practical question is that you are trying to address. How is the answer to this question important to the field, as this is not very clear.

Please, tell the readers what are the main findings from previous studies.

Explain the main reasons and gaps to justify your review. The authors just justify the article saying the absence of a previous review paper.

Also, finish with the main aim of the study.

Methods and Discussion

The topics are well written and well explored by the authors. Also, the tables are easy to understand.

Discussion

The authors claim to provide suggestions to test the cardiorespiratory system in water. The water temperature and level of immersion are two important points. However, the discussion still has a lack about the suggestions and some points needs to addressed:

How to evaluate the cardiorespiratory system during cycling and running underwater?

Should we follow the traditional protocols in the laboratory? Ramp test or step test? Should we perform lactate test or ventilatory thresholds? Should we use RPE, HR, VO2 system?

Is there a possibility to test the cardiorespiratory system using a constant test?

What are the practical implications about testing the cardiorespiratory system underwater?

Therefore, I understand the authors have to explore more the discussion and provide a detailed discussion to achieve consensus about cardiorespiratory tests underwater using a cycling and running exercise. 

Author Response

Thank you for the careful examination of our paper. We trust our explanations and additions to the paper sufficiently address your concerns and suggestions, and the revised manuscript aligns well with The International Journal of Environmental Research and Public Health publication goals.

This review investigated the different head-out water-based protocols used to assess cardiorespiratory fitness. I feel that this is an interesting paper about the possibility to use aquatic environment to perform tests and training for different populations. The study presented an overall experienced large number of papers with high quality and criteria. Also, there is no review about the theme. Therefore, while the manuscript was generally well written and fill a gap in the scientific literature, I believe that there are currently a number of issues that need to be addressed. Major concerns are also provided in a number of specific comments below.

Normally, the introduction has 4-5 paragraphs where the authors try to explain for the readers the background about theme. Please, provide more evidences.

More evidence and broader background were provided (lines 29-36).

It also needs to be clear what the practical question is that you are trying to address. How is the answer to this question important to the field, as this is not very clear.

The practical aspects and importance for the field were underlined (lines 41-51).

Please, tell the readers what are the main findings from previous studies.

This information was expanded (lines 38-41).

Explain the main reasons and gaps to justify your review. The authors just justify the article saying the absence of a previous review paper.

The review justification was better elaborated (lines 34-36 and 44-51).

Also, finish with the main aim of the study.

The aims of the study were clearly stated (lines 73-76). 

Methods and Discussion

The topics are well written and well explored by the authors. Also, the tables are easy to understand.

Discussion

The authors claim to provide suggestions to test the cardiorespiratory system in water. The water temperature and level of immersion are two important points. However, the discussion still has a lack about the suggestions and some points needs to addressed:

How to evaluate the cardiorespiratory system during cycling and running underwater?

The variety of the protocols conducted with special equipment does not allow to propose any recommendations for testing. The analyzed protocols are presented in details in table 3.

Should we follow the traditional protocols in the laboratory? Ramp test or step test? Should we perform lactate test or ventilatory thresholds? Should we use RPE, HR, VO2 system?

Although the review results do not give clear answers for the above questions, we addressed your questions directly related to the protocol in the discussion (lines 248-250) and conclusions (line 338-340). The definition of cardiorespiratory fitness assessment provided in the manuscript narrowed our analysis to VO2peak/VO2max. Taking into account that it was first attempt to review cardiorespiratory aquatic protocols, we did not intend to deeply analyze and give final recommendations about more physiological details (lactate test, RPE, HR etc.).

Is there a possibility to test the cardiorespiratory system using a constant test?

Unfortunately, our review did not give us any credibility to answer this question.

What are the practical implications about testing the cardiorespiratory system underwater?

The practical implications about testing the cardiorespiratory system underwater were described in the discussion section: water temperature (lines 252-262), water depth (lines 263-275), mode of exercise (lines 276-281) and exercise intensity (lines 300-307). These aspects were also underlined in the conclusion section (lines 334-338). 

Therefore, I understand the authors have to explore more the discussion and provide a detailed discussion to achieve consensus about cardiorespiratory tests underwater using a cycling and running exercise. 

Reviewer 2 Report

General Comments

The authors are commended for a well-written manuscript. However, there are significant concerns and issues with the manuscript in its current form that need to be addressed before being considered for publication. The theoretical framework of the study is apparently weak, and the research problem, despite the originality, is poorly relevant for both clinical and exercise-related settings (“no systematic review existed describing cardiorespiratory fitness protocols used in an aquatic environment”). It is the hope that the following critique will be received in the manner in which it is delivered and be used to improve the quality of the manuscript.

Major Concerns

MC1. Introduction – and throughout the manuscript. The authors are strongly encouraged to revise the text.

MC2. What is the relevance of this study? It is not made clear in your Introduction. Please re-visit.

MC3. The conclusions are weak and poorly relevant ("No single, broadly accepted head-out water-based protocol evaluating cardiorespiratory fitness arose"). Please re-visit.

Author Response

Thank you for the careful examination of our paper. We trust our explanations and additions to the paper sufficiently address your concerns and suggestions and the revised manuscript aligns well with The International Journal of Environmental Research and Public Health publication goals.

General Comments

The authors are commended for a well-written manuscript. However, there are significant concerns and issues with the manuscript in its current form that need to be addressed before being considered for publication. The theoretical framework of the study is apparently weak, and the research problem, despite the originality, is poorly relevant for both clinical and exercise-related settings (“no systematic review existed describing cardiorespiratory fitness protocols used in an aquatic environment”). It is the hope that the following critique will be received in the manner in which it is delivered and be used to improve the quality of the manuscript.

Major Concerns

MC1. Introduction – and throughout the manuscript. The authors are strongly encouraged to revise the text.

The introduction was revised. More evidence and broader background were provided (lines 29-36). The findings from previous studies were added (lines 38-41). The practical aspects and importance for the field were underlined (lines 41-51). The aims of the review were clearly stated (lines 73-76). 

MC2. What is the relevance of this study? It is not made clear in your Introduction. Please re-visit.

The relevance of this study was better elaborated in the introduction (lines 34-36 and 44-51).

MC3. The conclusions are weak and poorly relevant ("No single, broadly accepted head-out water-based protocol evaluating cardiorespiratory fitness arose"). Please re-visit.

We find this comment very general. Although the example is provided by Reviewer, it is still not clear why this conclusion is ‘weak’ and ‘poorly relevant’. In our opinion it is meaningful and gives readers information on the lack of consensus on standard head-out water-based protocol evaluating cardiorespiratory fitness. There are three more concrete conclusions on water temperature, depth and exercise intensity which give the practical implications about testing the cardiorespiratory system underwater. We also added suggestion about physiological parameters potentially relevant for cardiorespiratory aquatic protocols (lines 338-340).

Reviewer 3 Report

Really nice paper. Important content.

Abstract: It is unclear in the abstract whether you looked at 42 articles or 37 studies.

Methods: The methods are mapped out well. It is unclear, however, which criteria utilized by the two reviewers during their assessment. Listing these criteria would help. Are the criteria in the quality assessment section the same? If so, then state this. Avoid first-person language in line 68.

Figure 1. Excellent!

Line 91 should have a hyphen "Thirty-six"

Line 93 should have a hyphen "Cut-point"

Line 105 should have an apostrophe (papers')

Line 106 - Avoid first person language.

Tables 1 and 2 - Excellent!

Line 176 - Because 2 studies reported the same data, you might want to eliminate one from the study due to the redundancy. Were these two studies always "merged" throughout your paper? If not, then you should probably choose one, and eliminate the other from your review.

Line 197 - "DWR and DWE reported..." this wording is confusing. It looks this DWE and DWR and writing the paper. Recommend staying the the wording from the preceding paragraph. "For 12 protocols based on SWE and SWR, mean..."

Line 199, "Other outcome measures ... stated..." Again, it looks like the measure are writing the paper. Recommend something like this: "Other researchers of studies with outcome measures for DWE and DWR found that..."

In Line 174, it says, "Authors of 38 studies..." In line 209 is says, Twenty eight of 37 studies..." Why is there now 37 studies. Please clarify. Also, don't forget the hyphen in "Twenty-eight."

Line 235 - A table that summarizes this with recommended ranges for treatment intervention would be helpful here.

Line 243 - Participants who... instead of "participants that."

Line 255. Avoid first person language

Line 279 - Some really helpful information, here.

Author Response

Thank you for the careful examination of our paper. We trust our explanations and additions to the paper sufficiently address your concerns and suggestions and the revised manuscript aligns well with The International Journal of Environmental Research and Public Health publication goals.

Really nice paper. Important content. Thank you

Abstract: It is unclear in the abstract whether you looked at 42 articles or 37 studies.

We looked at 41 studies from which only 37 presented VO2peak/VO2max exact values. We added this information in the abstract (lines 17-18).

Methods: The methods are mapped out well. It is unclear, however, which criteria utilized by the two reviewers during their assessment. Listing these criteria would help. Are the criteria in the quality assessment section the same? If so, then state this.

The criteria used for inclusion of the studies are listed in the lines 94-97. The exclusion criteria are listed in the figure 1.

Avoid first-person language in line 68.

As the journal does not recommend which form to use we decided to use first person. The articles published in the International Journal of Environmental Research and Public Health use first-person language. Below is also the link which explains why we decided to use it. We leave it to the Editor if the change should be made.

https://oxfordediting.com/to-we-or-not-to-we-the-first-person-in-academic-writing/

Figure 1. Excellent! Thank you.

Line 91 should have a hyphen "Thirty-six" Corrected (line 118).

Line 93 should have a hyphen "Cut-point" Corrected (line 120).

Line 105 should have an apostrophe (papers') Corrected (line 131).

Line 106 - Avoid first person language. Explained in previous comment.

Tables 1 and 2 - Excellent! Thank you.

Line 176 - Because 2 studies reported the same data, you might want to eliminate one from the study due to the redundancy. Were these two studies always "merged" throughout your paper? If not, then you should probably choose one, and eliminate the other from your review.

Two studies presented the same data so they were merged as Brown et al., 1997. We eliminated one article from the total number analyzed as suggested the Reviewer. It was corrected in the abstract (line 15), results (lines 111-112, 118, 122, 124, 126, 127, 134, 139, 143, 201). We also added the information in the figure 1.

Line 197 - "DWR and DWE reported..." this wording is confusing. It looks this DWE and DWR and writing the paper. Recommend staying the the wording from the preceding paragraph. "For 12 protocols based on SWE and SWR, mean..."

Corrected (line 22).

Line 199, "Other outcome measures ... stated..." Again, it looks like the measure are writing the paper. Recommend something like this: "Other researchers of studies with outcome measures for DWE and DWR found that..."

Edited (lines 224-225).

In Line 174, it says, "Authors of 38 studies..." In line 209 is says, Twenty eight of 37 studies..." Why is there now 37 studies. Please clarify. Also, don't forget the hyphen in "Twenty-eight."

37 studies was the result of merging two articles in one Brown et al., 1997 which was explained in the first version of the manuscript in the lines 175-176. As suggested by the Reviewer we eliminated this one study during the eligibility assessment which was corrected in the article and explained in details in previous comment.

The hyphen was added (line 235).

Line 235 - A table that summarizes this with recommended ranges for treatment intervention would be helpful here.

Line 235 include the last part of the sentence “The recommended water temperature for cardiorespiratory fitness testing is 28-30C with difference of maximum 1C between testing participants, which the analyzed studies mainly incorporated.” We are not sure what the comment was about but the values of the water temperature are included in the tables with the protocols description.

Line 243 - Participants who... instead of "participants that." Corrected (line 270).

Line 255. Avoid first person language Explained in previous comment.

Line 279 - Some really helpful information, here. Thank you.

Round 2

Reviewer 1 Report

The authors have addressed the majority of my comments. Thank you. 

Author Response

The authors thank the Reviewer for the examination of our paper. 

Reviewer 2 Report

The authors are commended for a well-written revised manuscript. However, there are significant concerns and issues with the manuscript in its current form that need to be addressed before being considered for publication. I consider, therefore, the manuscript under review inadequate for publication. The study's theoretical framework is apparently weak, and the research problem, despite the originality, is poorly relevant for both clinical and exercise-related settings ("no systematic review existed describing cardiorespiratory fitness protocols used in an aquatic environment").

Author Response

Theoretical framework of our review is typical for systematic reviews. In our paper we delivered a clear and comprehensive overview of available evidence on cardiorespiratory fitness testing in water environment. We provided in the introduction findings from previous studies and explained the practical aspects and importance for the filed. There are no doubts that regular testing the individuals participating in aquatic program with a standardized cardiorespiratory protocols may give practitioners and clinicians very valuable information for establishing exercise guidelines, monitoring progress and making adjustments in both the intervention and training program. The theoretical framework of our study included identification of research gaps in current understanding of a field and highlighted methodological concerns in research studies that can be used to improve future work in the topic area.